# High-Definition Transcranial Direct Current Stimulation Improves Decision-Making Ability: A Study Based on EEG

**DOI:** 10.3390/brainsci13040640

**Published:** 2023-04-08

**Authors:** Yuwei Zhou, Guixian Xiao, Qing Chen, Yuyang Wang, Lu Wang, Chengjuan Xie, Kai Wang, Xingui Chen

**Affiliations:** 1Department of Neurology, The First Affiliated Hospital of Anhui Medical University, Hefei 230032, China; 2Department of Psychology and Sleep Medicine, The Second Affiliated Hospital of Anhui Medical University, Hefei 230601, China; 3Department of Neurosurgery, The Second Affiliated Hospital of Anhui Medical University, Hefei 230601, China; 4Department of Psychology, The School of Mental Health and Psychological Sciences, Anhui Medical University, Hefei 230032, China; 5Hefei Comprehensive National Science Center, Institute of Artificial Intelligence, Hefei 230088, China; 6Anhui Province Key Laboratory of Cognition and Neuropsychiatric Disorders, Hefei 230022, China

**Keywords:** decision-making, high-definition transcranial direct current stimulation (HD-tDCS), Iowa gambling task (IGT), event-related potentials (ERP), event-related spectral power (ERSP)

## Abstract

High-definition transcranial direct current stimulation (HD-tDCS) has been shown to modulate decision-making; however, the neurophysiological mechanisms underlying this effect remain unclear. To further explore the neurophysiological processes of decision-making modulated by HD-tDCS, health participants underwent ten anodal (*n* = 16)/sham (*n* = 17) HD-tDCS sessions targeting the left DLPFC. Iowa gambling task was performed simultaneously with electroencephalography (EEG) before and after HD-tDCS. Iowa gambling task performance, the P300 amplitude, and the power of theta oscillation as an index of decision-making were compared. Behavioral changes were found that showed anodal HD-tDCS could improve the decision-making function, in which participants could make more advantageous choices. The electrophysiological results showed that the P300 amplitude significantly increased in CZ, CPZ electrode placement site and theta oscillation power significantly activated in FCZ, CZ electrode placement site after anodal HD-tDCS. Significant positive correlations were observed between the changes in the percent use of negative feedback and the changes in theta oscillation power before and after anodal HD-tDCS. This study showed that HD-tDCS is a promising technology in improving decision-making and theta oscillation induced by may be a predictor of improved decision-making.

## 1. Introduction

In daily life, we make various decisions every day. Most of the time, the occurrence probability of selection outcomes is uncertain. This type of decision-making can benefit human survival and adaptation. Clinical studies have found that people with neurological and psychiatric disorders cannot make advantage choices under ambiguous condition. Therefore, it is essential to find an effective technological method to modulate decision-making ability.

Noninvasive brain stimulation techniques, such as transcranial magnetic stimulation (TMS) and transcranial direct current stimulation (tDCS), have been widely used to improve performance in several cognitive domains, including executive functions, memory performance, and decision-making in healthy subjects [1,2,3,4]. tDCS is a technique by applying weak electrical currents to the scalp to modulate the neuronal activity, and therefore change the behaviors [5,6]. Based on early experimental work, tDCS modulate cortical activity in a polarized manner at the neuronal level (anodal tDCS up-modulate the cortical activity and cathode down-modulate it) [7]. In addition, several studies have found that the regulatory effect of tDCS exceeds the applied time and is not spatially limited to the target region [8,9]. This may be attributed to the role with tDCS on synaptic plasticity as well as on brain networks. Converge form the previous studies, the result of applying tDCS to enhance decision-making were promising but also contradictory [10,11,12,13,14]. The size of electrode (conventional or high definition tDCS), current intensity, duration, session were the factors that influence the regulatory effect [15]. For example, a larger effect in multi-session anodal tDCS in cognition enhancement was found compared to single-session tDCS [16,17,18]. Similar result was also observed in a study with ten sessions anodal tDCS which showed improved executive control in individuals with a borderline personality disorder [19]. In addition, the conventional and high definition tDCS showed different effect on cognitive enhancement. The application of tDCS in neuropsychology establishes a causal relationship between modulated activity at the target region and resulting behavioral changes. Conventional tDCS produces diffuse current flow for the farther distance between the electrode montages which make it difficult to attribute the improved behavior to the increased activity in the target region [20]. High-definition transcranial direct current stimulation (HD-tDCS) can apply to brain regions with increased spatial focality and have longer-lasting regulatory effects compared to conventional tDCS [21,22]. Hence, we speculated that anodal HD-tDCS is a promising tool in improving decision-making.

In the present study, we focus on the left dorsolateral prefrontal cortex (DLPFC) given its established role in mediating ambiguous-risk decision-making [23,24]. Decision-making is a complex process which involved three stages: forming preferences, selecting and executing, and assessing outcomes [25]. Based on previous neuroimaging findings, the neural circuits involved in decision-making include DLPFC, ventral medial prefrontal cortex, orbital frontal cortex, and anterior cingulate cortex [26,27,28]. Among them, the DLPFC plays an important role in decision-making [25,29]. Specifically, it has a necessary role in the implementation of choice-induced preference change [30]. In addition, the DLPFC was found to be recruited in error monitoring to permit behavior to become adaptive. Learning and processing the outcomes to optimize behavior is also inseparable from the DLPFC [30,31]. Notably, the role of the DLPFC in decision-making is asymmetric. Patients with left DLPFC injury had decision-making impairment [32]. A tasked-related functional magnetic resonance imaging determined that adaptive decisions had a preference in left DLPFC but not right [33]. The modulation of left DLPFC through transcranial magnetic stimulation could result in making rational choices during decision-making task [33]. These studies suggested that the function of the left DLPFC was closely related to decision-making.

Iowa Gambling Task (IGT) is a common experimental paradigm for observing ambiguous decisions and has been proved to be useful in the detection of decision-making impairments in several neurological and psychiatric conditions [34,35]. Event-related functional magnetic resonance imaging studies showed that a neural circuitry involving the left DLPFC was activated during the IGT and that the changed activity of left DLPFC was associated with the IGT netscores [36]. Patients with DLPFC impairment had poor decision-making performance [37], especially when the damage is in the left side [17]. Therefore, IGT is a promising neuropsychological measurement paradigm for measuring decision-making function.

Most previous studies evaluating the effects of tDCS in the IGT used only behavioral outcomes [38,39], the modulation mechanism is still unclear. Hence, we adapted modified version of the IGT which was conducted during electroencephalograph (EEG) to obtain a thorough understanding of the neural basis of decision-making. Event-related potential (ERP) and event-related spectral power were adopted to measure the electrophysiological characteristics during IGT [40,41]. Previous ERP studies viewed P300 as a typical indicator to reflect rewarding processing in decision-making under uncertainty [42]. In a gambling task, trials with monetary losses elicited larger P300 amplitude than trials with monetary gains [43]. Moreover, it was found that P300 amplitudes seem to increase depending on the result of feedback, with strongest changes elicited by the losses [44]. The processing of reward and punishment stimuli in humans appears to involve brain oscillatory activity of several frequencies, probably each with a distinct function [41]. Theta power was found increased in mid-frontal areas after an error or negative feedback [45]. Moreover, theta oscillation appears to predict risk-taking behaviors during gambling tasks [46]. These studies suggested that electrophysiological indices including P300 and theta activity were promising candidates for decision-making processing. To our knowledge, the current study is the first one that combines behavioral and electrophysiological data (ERPs) to explore the electrophysiological mechanisms of decision modulation by anodal HD-tDCS during IGT.

The objective of present study was to explore the effect of anodal HD-tDCS in improving the decision-making of healthy individuals and explore the neural mechanisms related to decision-making. The following assumptions were made: (1) Anodal HD-tDCS over left DLPFC can effectively improve the decision-making performance; (2) improved performance was associated with the changes of P300 amplitude and the power of theta oscillation by HD-tDCS.

## 2. Materials and Methods

### 2.1. The Procedure

The current experiment was designed to be randomized and double-blind. In this study, the participants and investigators who guided the EEG were blinded to group information. Only the technicians performing the tDCS procedure and the investigator assigning the randomization without patient contact were aware of the group information. Participants who meet inclusion criteria were randomly assigned in active group and sham group according to a computer-generated list.

As shown in Figure 1B, 30 min of 2 mA HD-tDCS was delivered daily to the participants consecutively for 10 days. EEG recording combined with IGT and Montreal Cognitive Assessment (MoCA) was carried out one day before the first HD-tDCS and one day after the last HD-tDCS session.

### 2.2. Participants

Thirty-six individuals with Han Chinese who have high school degree or above were recruited for this research study. They were randomized into the active and sham groups (*n* = 18 in each group). Three subjects were excluded due to fail receiving the daily HD-tDCS for 10 consecutive days. In the end, thirty-three participants were assigned in the active group (*n* = 16, 9 females) and the sham group (*n* = 17, 7 females). As shown in Table 1, there were no significant differences between the two groups in gender (χ^2^ = 0.750, *p* = 0.387), age (z = −1.300, *p* = 0.194), or education level (z = −0.112, *p* = 0.911). Every participant needed to complete a questionnaire by interview before taking part in the study. All participants meet the inclusion criteria with no brain injury, mental illness, neurological illness, or drug/alcohol dependence. All participants had normal or corrected-to-normal vision. In addition, none of the subjects participated in tES or TMS studies. Each of them received RMB 600 after completing the experiment. The procedures of this study were approved by the ethics committee of Anhui Medical University. Each participant has signed an informed consent form before the study.

### 2.3. Task

We employed a computerized version of the IGT for ERP recordings [34]. At the beginning of the IGT, all participants were told that they would complete a gambling game and have a stake of RMB 1000. When the experiment started, two-choice stimulus with different number was presented in the screen, a 50-point bet (“F” key) and a 100-point bet (“J” key), which represented the monetary value in RMB. The 50-point bet is a low-risk choice and 100-point bet is a high-risk choice, because of the bet of 50-point with 40 percent loss probabilities and the bet of 100-point with 60 percent loss probabilities. At each trial, when the subject made their choices, the screen had a plus of 200–400 ms seconds, followed by a cartoon face with smiling face (winning) or crying of face (losing). The cartoon face lasting for 1000 ms. The computer then displayed a line of text and numbers, informed participants the result of the selection, and activated the starting interface one second later. The entire task was consisted of 300 trials which took about 15 min to complete it. They were also informed to win as much money as possible based on the results of each feedback. The overall experiment was divided into three segments, where all of the winning or losing situations during the whole segment were displayed on the monitor screen at the end of every segment. The task procedure was showed in Figure 2. In addition, “high-risk loss” was defined as when participants chose a high-risk choice and suffered a loss. A “low-risk win” was defined as when participants chose low-risk choice and profited. As a result, the percent use of negative feedback was calculated by dividing the number of times the participant chose the low-risk choice after receiving the high-risk loss by the number of times the participant received the high-risk loss. The percent use of positive feedback was defined as the number of times a participant chose low-risk choice again after experiencing a low-risk win divided by the number of times the participant experienced a low-risk win.

### 2.4. High-Definition Transcranial Direct Current Stimulation

According to the foregoing, left DLPFC is pivotal in IGT. In the present study, a battery-driven constant current stimulator (Neuroelectrics, Barcelona, Spain) was used to deliver current. Five Ag-AgCl with a diameter of 12 mm sintered ring electrodes were held in plastic casings filled with conductive gel, embedded in an EEG cap, and attached to the adaptor device. Based on the 10–20 EEG system, the central electrode as anodal electrode was located in F3 (2 mA), and the remaining four return electrodes as cathodal electrode were located in F1/F5/FC3/AF3 (0.5 mA each) [6]. Sham HD-tDCS used the same electrode montage as the anodal stimulation. Anodal HD-tDCS was applied at 2 mA for 30 min, with 30 s ramped up and 30 s ramped down. Previous studies have shown that such intensity and total charge were safe for humans and sufficient to influence DLPFC activities [47,48]. The sham HD-tDCS stimulation duration was 60 s, with current fading in and out, which was similar to that of anodal HD-tDCS. The sham group subjects had the same HD-tDCS regimen and duration as those of the active group, while no current was given. During every session, all participants were asked to close their eyes to reduce individual difference. As shown in Figure 1A, the montage was precisely chosen to target the area to the left of the DLPFC, and this location was confirmed by the electrical field distribution simulation.

### 2.5. Electrophysiological Recording

EEG recording was tested in a sound-resistant chamber with a comfortable chair, and the temperature was kept suitable. In addition, all participants were asked not to consume any coffee or alcoholic beverages 2 h before the experiment. And all participants were asked to wash their hair and keep their scalp dry. None of the participants was sleepy or fatigue during the EEG recording. A scalp elastic cap of 64 electrodes was placed according to the International 10–20 system and used to record EEG data (Neuro Scan, Sterling, VA, USA). During the recording, an electrode located at the forehead was used as the ground, and all EEG channels were referenced to the left mastoid [49]. The electrical signals of the eye were collected to determine the eye-open blink status. The horizontal electrooculogram was attached one centimeter on the outside of both the lateral eye fissures. The vertical electrooculogram was placed above the midpoint of the left eyebrow and 1 cm below the lower eyelid. All electrodes’ scalp resistance were kept below 5 kΩ. The collected EEG data were displayed on another computer via an amplifier. The amplifier was set bandpass filtered at 0.1–100 Hz and sampled continuously at 1000 Hz. Then, the stored data were analyzed by MATLAB software (R2019b, The MathWorks Inc., Natick, MA, USA) and the EEGLAB toolbox.

To acquire cleaned signals, we downsampled the raw signals to 500 Hz and analyzed with a bandpass at 0.1–30 Hz. The offline data were referenced to the average of the left and right mastoids. To eliminate the effects of electrooculogram and movement artifacts, trials were excluded if the signals exceeded ±100 μv. To obtain cleaned data, blinks, eye movements, electromyography, and other artifact-independent components were removed through Independent component analysis which performed using the EEGLAB toolbox. Before HD-tDCS, the mean rejected epochs was 3.47 [95%, (2.68, 4.27)] in active group, 3.56 [95%, (2.57, 4.55)] in sham group. After HD-tDCS, the mean rejected epochs were 3.76 [95%, (2.79, 4.74)] in active group and 3.50 [95%, (2.51, 4.49)] in sham group. Rejection epochs did not differ significantly among groups (F = 0.482, *p* = 0.493).

The cleaned ERP waveform was made up of 1200 ms epochs. Each epoch included 200 ms before the onset of cartoon face and 1000 ms after the cartoon face. Converging previous studies related to P300 component, the midline sites (FZ, CZ, CPZ) showed larger P300 amplitude [50,51]. Based on the grand average time-frequency plots, we chose the interest of frequency window between 4 and 6 Hz and the time window between 350 and 450 ms for the maximal power of theta oscillation was centered on FCZ site, which was consistent with the previous studies [52].

### 2.6. Statistical Analysis

All statistical analyses were performed using SPSS (version 16; Chicago, IL, USA). Normality of the data distribution was assessed using the Shapiro–Wilk tests. Baseline comparisons between the groups were performed using two independent sample t-tests (data with normal distribution) or Mann–Whitney U test (data with non-normal distribution). Sex comparisons were made using the chi-square test. To determine the behavioral results before and after HD-tDCS, we defined the total netscores as the difference between the number of low-risk choice and high-risk choice. Two-factor repeated-measures analysis of variance (ANOVA) was performed to compare the difference in total netscores, money accounts, the percent use of positive feedback, the percent use of negative feedback between the two groups before and after HD-tDCS. In addition, the 300 trials were divided into five equal block. The netscores of each block were calculated to investigate whether decision-making changed during the task. Time points (pre-tDCS and post-tDCS) and block were used as within-subject factors, and group was used as a between-subject factor. The average amplitude of P300 and the power of theta-band oscillation were analyzed by multivariate repeated-measures ANOVAs. There were four within-subject factors, including feedback type (lose and win), intensity (50 as low-risk choice and 100 as high-risk choice), electrode (FCZ, CZ, CPZ), and time points (pre-tDCS and post-tDCS); one between-subject factor was group (active group and sham group). The degrees of freedom of the F-ratios were adjusted according to the Greenhouse–Geisser epsilon correction in all analyses. Effect sizes were calculated as partial eta squared (η2p). Partial eta squared(η2p) values were obtained to examine the sizes of effects in the ANOVA models, where 0.05 indicated a small effect, 0.1 indicated a medium effect, and 0.2 indicated a large effect [53]. The Bonferroni method was used to correct multiple comparisons in post hoc tests. Pearson’s correlation coefficients were calculated to examine the strengths of the associations between the changes of behavior performance and changes of P300 amplitude, as well as theta oscillation. *p* < 0.05 represents statistical significance. A Chi square test was used to assess the blinding efficacy of the procedure.

## 3. Results

### 3.1. Demographic and Behavioral Performance

As shown in Figure 3B, two-way repeated-measures ANOVA was used to compare the interaction between time and groups to examine IGT performance, including netscores [F(1, 31) = 5.036, *p* = 0.032, η2p = 0.140], money accounts [F(1, 31) = 6.174, *p* = 0.01, η2p = 0.166], the percent use of positive feedback [F(1, 31) = 1.886, *p* = 0.180, η2p = 0.059] and the percent use of negative feedback [F(1, 31) = 5.620, *p* = 0.024, η2p = 0.153]. For a further analysis of the simple effects of time, we found that in the active group, the money accounts [F(1, 31) = 5.692, *p* = 0.023, η2p = 0.155], the IGT total netscores [F(1, 31) = 7.954, *p* = 0.008, η2p = 0.204], the percent use of positive feedback [F(1, 31) = 6.597, *p* = 0.015, η2p = 0.175], and the percent use of negative feedback [F(1, 31) = 6.043, *p* = 0.020, η2p = 0.163] increased over time. In addition, we also compared the netscores of the two groups from five blocks of the two groups before and after HD-tDCS. The interaction effects between time points and group [F(1, 31) = 3.624, *p* = 0.066, η2p = 0.105], block and group [F(4, 28) = 1.071, *p* = 0.309, η2p = 0.033], and block and time points [F(4, 28) = 2.371, *p* = 0.134, η2p = 0.071] were not significant. The interaction effects among time points, block, and group [F(4, 28) = 1.518, *p* = 0.201, η2p = 0.047] were not significant. We further analyzed the main effect of time and block, the net scores of both groups showed an increasing trend as the task progressed, as shown in Figure 3A. In the active group, the netscores increased significantly on block1 and block4 after HD-tDCS (all *p* < 0.05).

### 3.2. P300 Amplitude

Based on the overall average ERP evoked by cartoon face, we set the time window of P300 as 350–450 ms, which is also consistent with the definition of the P300 time window in the previous literature [51]. Multivariate repeated-measures ANOVA was performed to explore the changes in the mean P300 amplitude before and after HD-tDCS. Intensity, feedback type, electrode, and time were the within-subject factors, and group was the between-subject factor. We found a significant main effect of intensity [F(1, 31) = 46.776, *p* < 0.001, η2p = 0.601] and time [F(1, 31) = 4.499, *p* = 0.042, η2p = 0.127] and a marginal main effect of feedback type [F(1, 31) = 3.426, *p* = 0.074, η2p = 0.100] and channel [F(1, 31) = 3.906, *p* = 0.057, η2p = 0.127]. The CZ electrode under high-risk loss conditions had the largest amplitude. A mixed-design two-way ANOVA indicated a significant group × time (2 × 2) interaction effect for theta frequency neural oscillation activity in three electrodes under loss and win conditions. In Table 2, all electrodes had a significant interaction between time and group under the high-risk loss condition [FCZ: F(1, 31) = 4.925, *p* = 0.034, η2p = 0.137; CZ: F(1, 31) = 5.460, *p* = 0.026, η2p = 0.150; CPZ: F(1, 31) = 5.525, *p* = 0.025, η2p = 0.151]. For a further analysis of the simple effect of time and stimulation, significant difference were found in CZ [F(1, 31) = 6.697, *p* = 0.015, η2p = 0.178], CPZ [F(1, 31) = 5.295, *p* = 0.028, η2p = 0.146] electrodes under high-risk loss condition in active group. As shown in Figure 4, this simple analysis of effects revealed that in CZ, CPZ electrodes site, P300 amplitude changes largely under the high-risk loss condition in the active group than sham group.

### 3.3. Theta Oscillations

Time-frequency analysis was performed to explore the neural oscillation in decision-making. To further explore the electrophysiological mechanisms of decision-making, we focused on the time window of 350–450 ms after the cartoon face presented. We found that neural oscillation at 4–6 Hz in the active group was significantly activated. In Figure 5A, significant main effects were found on feedback [F = 29.611, *p* < 0.001, η2p = 0.489], intensity [F = 9.341, *p* = 0.005, η2p = 0.232], electrode [F = 6.430, *p* = 0.016, η2p = 0.173], and time [F = 11.435, *p* = 0.002, η2p = 0.269]. Loss condition induced a higher power than winning, selected 100-induced power above 50, FCZ site had the largest power, the power was significantly higher than before HD-tDCS. The results showed that under high-risk loss conditions, the activation of the FCZ was more obvious. As with the analysis method of P300, the interaction effect of time and group under the loss and win conditions is shown in Figure 5B [FCZ: F(1, 31) = 4.026, *p* = 0.054, η2p = 0.115; CZ: F(1, 31) = 5.700, *p* = 0.023, η2p = 0.155; CPZ: F(1, 31) = 3.853, *p* = 0.059, η2p = 0.111). Detailed results are presented in Table 3. Similarly, the simple effect of time and stimulation analysis shown significant difference in FCZ [F(1, 31) = 4.695, *p* = 0.038, η2p = 0.132], CZ [F(1, 31) = 4.493, *p* = 0.042, η2p = 0.146] electrodes under high-risk loss condition in active group. As shown in Figure 5, this simple analysis of effects revealed that in FCZ, CZ electrodes site, theta oscillation activated more obviously under the high-risk loss condition in the active group than sham group.

### 3.4. Correlation Analysis

To further explore the electrophysiology mechanism of decision-making, we analyzed the correlation between the changes of IGT performance and the changes of decision-making electrophysiology components before and after HD-tDCS. As shown in Figure 6, the changes in the percent use of negative feedback showed positive correlations with the changes of theta oscillation power in FCZ (r = 0.568, *p* = 0.011), and CZ (r = 0.501, *p* = 0.024) sites, but not with IGT netscores, money accounts. Following anodal HD-tDCS modulation, the subjects switched their behavior, and selected low-risk choices after high-risk loss. There were no significant correlations between the changes in IGT netscores, money accounts, or the percent use of negative feedback and the changes in P300 amplitude (all *p*’s > 0.05). The correlation analysis within sham group indicated that the changes in the IGT performance were not correlated with the changes in P300 amplitude or theta oscillation power after HD-tDCS (all *p*’s > 0.05). There were no correlations in the changes of the percent use of positive feedback with the changes of P300 amplitude or theta oscillation power. In addition, there was no significant correlation between the decision-making performance and the MoCA score.

### 3.5. Additional Analyses

There is evidence that gender is involved in the decision-making enhancement by tDCS stimulation over DLPFC [54]. A multivariate ANOVAs was performed to determine whether gender had an influence on the decision-making performance and electrophysiological characteristics elicited in response to the IGT before and after HD-tDCS. No significant effects of gender were found on behavioral and electrophysiological characteristics before HD-tDCS. Furthermore, there were no main effect of gender or interaction effect of gender and group on decision-making. Repeated-measures ANOVA showed that there were no gender-dependent behavioral and electrophysiological differences on effects of HD-tDCS in the IGT (all *p*’s > 0.05).

### 3.6. Feasibility and Blinding Efficacy of the Procedure

None of the participants experienced serious adverse effects during or after HD-tDCS. Five in thirty-three participants reported tingling and burning sensation, 3 in 33 participants reported itching sensation under the electrodes during the first 30 s of stimulation. Subjects in the sham group reported the same initial sensation. Furthermore, at the end of HD-tDCS, all participants include three subjects who did not complete the experiment were asked if they could guess what type of HD-tDCS they received. We suggest that our blinding procedure was successful for participants were unable to distinguish between active and sham stimulation (active group = 56.25 %; sham group = 41.17 %; χ^2^ = 1.273, *p* = 0.529).

## 4. Discussion

In the present study, we explored the modulate effects of HD-tDCS on the behavior and the potential electrophysiological mechanism of the effect. We found participants have better decision-making performance after anodal HD-tDCS. Furthermore, we illuminate the electrophysiological mechanisms underlying these behavioral effects of tDCS on decision-making by EEG technology. Specifically, anodal tDCS applied left DLPFC induced the change in decision-making neural activity through P300 amplitude and the theta oscillation power.

Our results replicated studies which show that tDCS can increase IGT performance in decision-making. Related cognitive and neural mechanisms may be explain the decision-making enhancement of HD-tDCS. Decision-making is a component of cognitive control that plays a central role in modulating behavioral responses in a goal-directed manner [1]. The left DLPFC is the main region involved in cognitive control, which coordinates the reward processing system and the risk processing system [25]. Neuroimaging studies demonstrated that better strategy decision was associated with left DLPFC activity [29]. It is therefore possible that the up regulation of the left DLPFC enabled better decision-making. As we all know, tDCS can increase cortical excitability in targeted region. Thus, tDCS induced improved performance of decision-making which may be contributed to the increased activity of left DLPFC. In addition to promoting the direct activity of the left DLPFC, it is worth to note the indirect modulation of the relevant regions, such as ventral medial prefrontal cortex, orbital frontal cortex along with anterior cingulate cortex via their dense anatomic connections [4,55]. In the present study, participants adjust briefly after punishments to select the low-risk choice more frequently which may be contributed to the activated amygdala circuit [56]. And participants select low-risk again after reward which is relevant to activated ventral striatal circuit. Both circuits were mediated by left DLPFC [57]. Therefore, the circuit of decision-making were activated which can induced optimized decision behavior. In addition, recent studies suggested that a wide range of cortical and subcortical brain regions was modulated by reward expectation [58], left DLPFC play a pivotal role in neuromodulation specially [29]. Based on this standpoint, we suggested that the mechanism of tDCS effects is through altering brain network connectivity. Interesting, a study combined tDCS with fMRI suggested that tDCS over DLPFC alters connectivity in cortical and subcortical reward systems [59]. Therefore, the improved decision-making performance was induced by tDCS over the left DLPFC and the function connectivity of related brain regions which were involved in decision-making.

As mentioned before, electrophysiological feature is important in the present study to explore the underlying mechanism of the modulate effect. P300 was elicited during behavior adjustment which may be associated with frontal dopaminergic (DA) activity. A number of neurological and psychiatric studies had demonstrated that DA system is involved in the reward processing [60,61]. Furthermore, the level of DA was increased after tDCS applied the DLPFC [62]. Therefore, the tDCS effects on neurotransmitters might explain the changes in electrophysiological features, given that P300 is associated with frontal DA activity. In line with the previous study, tDCS over frontal region significantly increases P300 amplitude during different tasks [63]. In the present study, the P300 amplitude increased after tDCS over left DLPFC, which may be through changing the DA level. According to previous studies, P300 is indeed an important component of the decision-making task, and P300 is sensitive to the magnitude of reward, with a more positive response to a larger (whether positive or negative) than to a smaller reward [64]. However, the changed amplitude of P300 induced by tDCS could not explain the improved decision-making performance. One study suggested that early negative wave, P200, feedback-related negativity, P300 and late positive potential were all elicited during IGT [43]. In future, we can explore the other ERP components to better understand the electrophysiology mechanism of the modulation effect in this study.

Serving as a method to measure neural oscillations, event-related spectral power can not only supplement traditional ERP analysis but also independently visualize the spectral power of the average ERP over a wide band range over time [65,66]. Previous studies suggested that the processing of feedback information involves several distinct processes, which are subserved by oscillations of different frequencies [41]. Theta oscillation was a prominent time-frequency component implicated during decision task which was widely used in nervous and psychiatric disorders, such as autism spectrum disorder, Parkinson’s disease, schizophrenia [35,67,68]. Numerous electrophysiologic literature suggested this reward-related theta activity may be generated by frontal cortex, while anterior cingulate cortex and orbitofrontal cortex were possible contributions of brain regions [46,69,70]. In fact, except activated targeted region, tDCS can also alter the activation and functional connectivity in regions distal to the electrodes. Therefore, tDCS over left DLPFC may change the connectivity of task-related brain regions to modulate the theta oscillation. Similarly, the power of theta oscillation was increased after tDCS was applied to left DLPFC in the present findings. Moreover, theta oscillation was more sensitive to negative feedback [71], which is consistent with the present finding that the changes in the percent use of negative feedback were positively correlated to the increased power of theta oscillation. It is possible to speculate that theta oscillation was involved in neuromodulation mechanism of decision-making.

Several limitations should be mentioned regarding the present study. First, the samples were relatively small, and the results may be affected by individual heterogeneity. Additionally, gender difference in information processing [72], sensitive to tDCS in decision-making has been observed in previous studies [73]. We found no significant gender difference in this study. The possible reason may be the relatively small sample. Therefore, a large-scale study that focuses on the gender is needed in the future. Second, the neural mechanisms of decision-making should be further validated by a variety of technologies. In the future, we can combine magnetic resonance imaging to explore the circuits between brain regions and further explain the neural mechanism of decision-making. Third, we did not explore the lasting time of the improved decision-making effects in the current study. Longitudinal studies are necessary to further explore the lasting effects of HD-tDCS on the decision-making.

## 5. Conclusions

In the present study, we demonstrated the modulation effect of anodal HD-tDCS in decision-making. The improved decision-making performance was associated with the increased power of theta oscillation. Furthermore, this study supports that theta oscillation may be involved in neuromodulation mechanism of decision-making. These findings support that HD-tDCS is an appropriate protocol for facilitating the decision-making process for those who are diagnosed with decision-making dysfunction disorder. In addition, the approach of tDCS combined with EEG provides a direction for exploring the electrophysiological mechanisms in cognition.

## Figures and Tables

**Figure 1 brainsci-13-00640-f001:**
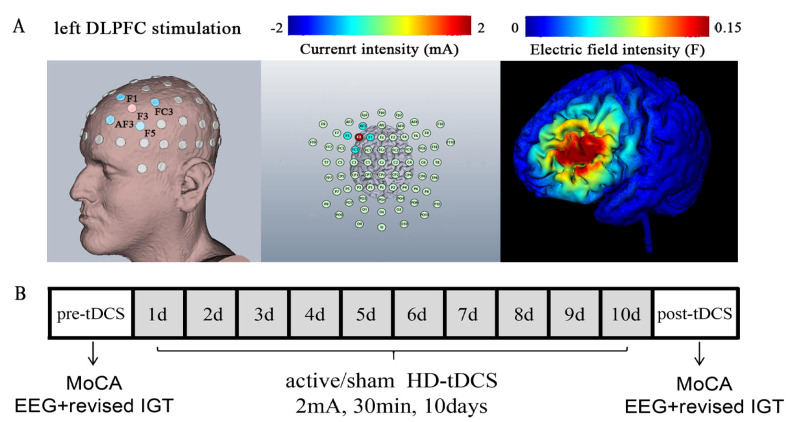
The experimental protocol and high-definition neuromodulation guided by model. (**A**) shows the study was completed within 12 days. Ten consecutive days for HD-tDCS sessions, two days for EEG recording and cognitive assessment which occur before HD-tDCS and after HD-tDCS. (**B**) shows the three-dimensional reconstructions of cortical of the electrical field models and the left DLPFC HD-tDCS protocols. The left DLPFC protocol included (in mA): F3 (2.0), AF3(−0.5), F1 (−0.5), F5 (−0.5), FC3 (−0.5).

**Figure 2 brainsci-13-00640-f002:**
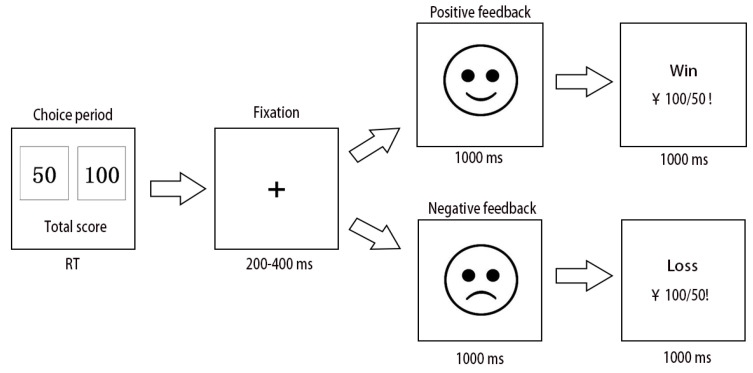
Illustration of the revised Iowa gambling task (IGT). There were two keys corresponding to two choices, and participants were asked to make 300 choices. After participants made a button response, it was followed by a 200–400 ms fixation point. After the fixation, a cartoon face persisted as feedback indicating whether they lost or won in the trial. Subsequently, a numerical stimulus popped up on the computer screen to indicate the selected consequence, which lasted for 1000 ms.

**Figure 3 brainsci-13-00640-f003:**
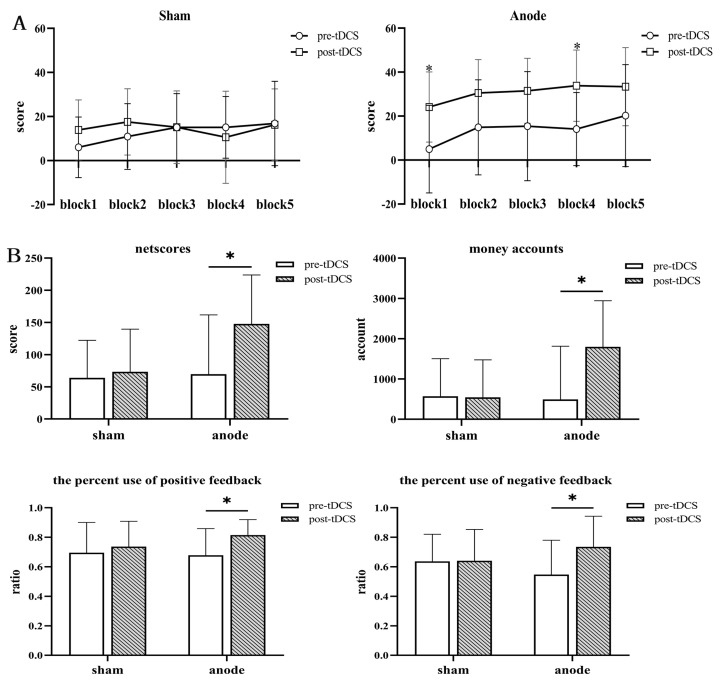
Performance of the active and sham groups in the revised Iowa Gambling Test (IGT). Comparisons of the two groups’ netscores (the number of low-risk choice minus the number of high-risk choice) on the six blocks (**A**), final total netscores, final monetary amount, the percent use of positive feedback, and the percent use of negative feedback (**B**) before and after treatment [bars indicate the standard error of the mean. * *p* < 0.05].

**Figure 4 brainsci-13-00640-f004:**
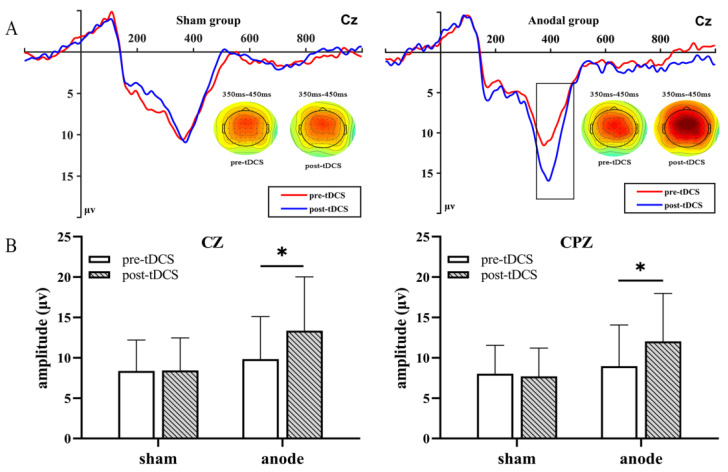
The results obtained from the CZ electrode are shown in (**A**). A significant difference in P300 was seen before and after treatment under high-risk conditions based on a paired *t* test. The black boxes define the time windows of interest where the amplitude increases significantly. In addition, (**B**) shows the P300 amplitude increased bar graph at two locations [bars indicate the standard error of the mean. * *p* <0.05].

**Figure 5 brainsci-13-00640-f005:**
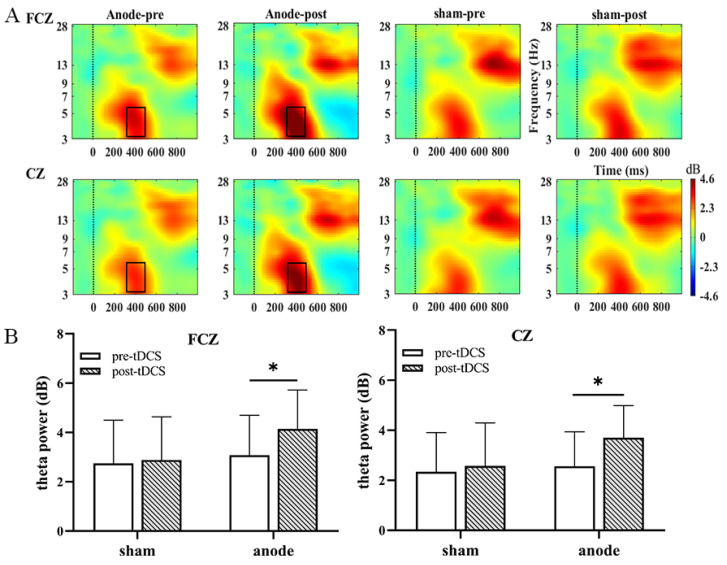
The results obtained from the FCZ and CZ electrodes are shown in (**A**). A significant difference in ERSP is seen between the two conditions (winning and losing) before and after treatment under high-risk conditions based on a paired *t* test. The black boxes define the time-frequency region of interest where the power increases significantly. In addition, (**B**) shows the theta oscillation active bar graph at two locations [bars indicate the standard error of the mean. * *p* <0.05].

**Figure 6 brainsci-13-00640-f006:**
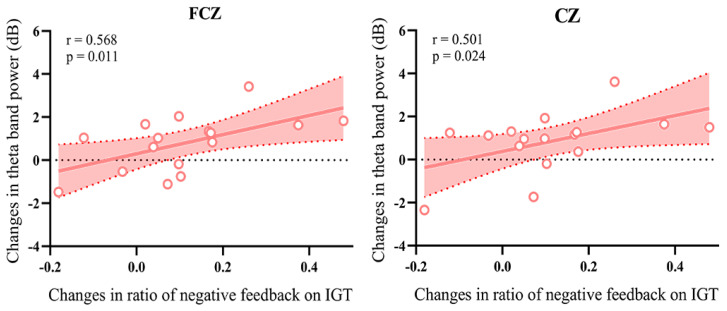
Correlation of changes between IGT performance and theta oscillation power. The increased power values of theta oscillation were significantly positively correlated with the changes in the percent use of negative feedback after HD-tDCS [The red shaded areas are the 95% confidence intervals. The circles represented the value of each participants in active group].

**Table 1 brainsci-13-00640-t001:** Demographic characteristics and IGT performance of participants.

	Active Group (*n* = 16) Mean (SD)		Sham Group (*n* = 17) Mean (SD)		Baseline Comparison	
	Pre	Post	Pre	Post	χ^2^/z/t	*p*
Age (years)	23.31 (3.26)	-	21.88 (2.26)	-	−1.300 ^b^	0.194
Education(years)	15.81 (1.94)	-	15.76 (1.86)	-	−0.112 ^b^	0.911
MoCA	28.56 (0.96)	28.94 (1.06)	28.76 (1.03)	28.82 (1.19)	−0.601 ^b^	0.548
Sex(male/female)	7/9	-	10/7	-	0.750 ^a^	0.387
Total netscores	69.63 (92.13)	153.38 (72.64)	63.88 (58.52)	73.41 (66.31)	0.215 ^c^	0.831
Money accounts	493.75 (1320.09)	1796.88 (1145.78)	570.59 (934.25)	544.12 (933.05)	−0.194 ^c^	0.847
Ratio of positive feedback	0.68 (0.18)	0.81 (0.11)	0.70 (0.20)	0.74 (0.17)	−0.259 ^c^	0.798
Ratio of negative feedback	0.55 (0.23)	0.73 (0.21)	0.64 (0.18)	0.64 (0.21)	−1.228 ^c^	0.229

Note: ^a^ Chi-squared test. ^b^ Mann-Whitney U test. ^c^ two independent sample *t*-test. Abbreviations: IGT, Iowa gambling task; SD, standard deviation; MoCA, Montreal Cognitive Assessment.

**Table 2 brainsci-13-00640-t002:** Planned 2 × 2 repeated measure ANOVA on P300 amplitude over time (pre-tDCS, post-tDCS).

	Active Group (*n* = 16, 7 Males) Mean (SD)		Sham Group (*n* = 17, 10 Males) Mean (SD)		Factor Time		Effect Size	Group by Time Interaction		Effect Size
	Pre	Post	Pre	Post	F	*p* ^a^		F	*p* ^a^	
**P300 amplitude**										
FCZ100	10.08 (5.90)	14.04 (7.15)	8.90 (4.65)	9.19 (4.80)	6.581	*0.015* *	0.175	4.925	0.034 *	0.137
FCZ101	9.51 (4.37)	11.12 (5.17)	8.07 (5.07)	9.32 (4.57)	3.355	*0.077*	0.098	0.052	*0.822*	0.002
FCZ50	8.80 (5.91)	10.51 (8.06)	6.76 (5.14)	8.92 (5.01)	6.050	*0.020* *	0.163	0.082	*0.776*	0.003
FCZ51	6.73 (3.68)	7.46 (4.46)	6.01 (4.04)	6.40 (4.18)	0.829	*0.370*	0.026	0.073	*0.789*	0.002
CZ100	9.84 (5.25)	13.36 (6.67)	8.36 (3.84)	8.43 (4.03)	5.910	*0.021* *	0.160	5.460	*0.026* *	0.150
CZ101	9.98 (4.36)	11.17 (4.85)	8.59 (4.92)	9.49 (4.83)	1.756	0.195	0.054	0.035	0.854	0.001
CZ50	8.37 (5.67)	9.82 (8.03)	6.45 (4.59)	8.20 (4.58)	4.224	*0.048* *	0.120	0.036	0.850	0.001
CZ51	7.17 (3.78)	7.61 (4.40)	6.28 (3.99)	6.28 (4.05)	0.147	0.704	0.005	0.148	0.703	0.005
CPZ100	8.97 (5.10)	12.04 (5.94)	8.02 (3.52)	7.70 (3.49)	3.636	0.066	0.105	5.525	*0.025* *	0.151
CPZ101	9.34 (4.29)	10.80 (4.82)	8.74 (4.32)	9.42 (4.87)	2.035	0.164	0.062	0.265	0.610	0.008
CPZ50	7.44 (5.95)	9.02 (6.77)	6.42 (4.26)	7.11 (4.04)	2.571	0.119	0.077	0.389	0.538	0.012
CPZ51	6.95 (4.29)	7.75 (4.26)	6.60 (3.77)	6.08 (3.76)	0.065	0.801	0.002	1.422	0.242	0.044

Note: FCZ100, CZ100, CPZ100: loss100 condition in FCZ, CZ, CPZ electrodes. FCZ101, CZ101, CPZ101: win100 condition in FCZ, CZ, CPZ electrodes. FCZ50, CZ50, CPZ50: loss50 condition in FCZ, CZ, CPZ electrodes. FCZ51, CZ51, CPZ51: win100 condition in FCZ, CZ, CPZ electrodes. ^a^ Bonferroni method: adjustments to post hoc multiple comparisons. * *p* < 0.05. Abbreviations: ANOVA, repeated-measures analysis of variance; SD, standard deviation.

**Table 3 brainsci-13-00640-t003:** Planned 2 × 2 repeated measure ANOVA on theta oscillation power over time (pre-tDCS, post-tDCS).

	Active Group (*n* = 16, 7 Males) Mean (SD)		Sham Group (*n* = 17, 10 Males) Mean (SD)		Factor Time		Effect Size	Group by Time Interaction		Effect Size
	Pre	Post	Pre	Post	F	*p* ^a^		F	*p* ^a^	
**Theta oscillation power**										
FCZ100	3.08 (1.62)	4.14 (1.58)	2.74 (1.75)	2.88 (1.75)	6.797	0.014 *	0.180	4.026	0.054	0.115
FCZ101	1.18 (1.53)	1.61 (2.28)	1.17 (1.58)	1.76 (1.41)	2.638	0.114	0.078	0.060	0.808	0.002
FCZ50	2.49 (2.09)	3.43 (2.11)	2.30 (1.65)	2.30 (1.99)	2.819	0.103	0.083	2.902	0.098	0.086
FCZ51	1.08 (1.43)	1.47 (1.13)	0.59 (0.58)	1.10 (1.37)	5.956	0.021 *	0.161	0.112	0.740	0.004
CZ100	2.56 (1.37)	3.70 (1.28)	2.34 (1.56)	2.57 (1.72)	13.227	0.001 **	0.300	5.700	0.023 *	0.155
CZ101	1.49 (1.57)	1.73 (2.48)	1.51 (1.85)	1.97 (1.46)	1.093	0.304	0.034	0.103	0.750	0.003
CZ50	2.13 (1.89)	3.11 (2.02)	1.96 (1.37)	2.30 (1.99)	5.290	0.028 *	0.146	1.268	0.269	0.039
CZ51	1.24 (1.48)	1.59 (1.10)	0.90 (0.82)	1.12 (1.33)	2.702	0.110	0.080	0.144	0.707	0.005
CPZ100	1.96 (1.57)	3.06 (1.29)	1.93 (1.60)	2.09 (1.49)	7.054	0.012 *	0.185	3.853	0.059	0.111
CPZ101	1.67 (1.48)	2.10 (2.29)	1.56 (1.86)	2.16 (1.54)	2.792	0.105	0.083	0.071	0.792	0.002
CPZ50	1.81 (1.70)	2.62 (1.74)	1.62 (1.24)	1.74 (1.71)	2.764	0.106	0.082	1.504	0.229	0.046
CPZ51	1.13 (1.46)	1.77 (1.11)	1.09 (0.95)	1.34 (1.29)	7.974	0.008 **	0.205	1.495	0.231	0.046

Note: FCZ100, CZ100, CPZ100: loss100 condition in FCZ, CZ, CPZ electrodes. FCZ101, CZ101, CPZ101: win100 condition in FCZ, CZ, CPZ electrodes. FCZ50, CZ50, CPZ50: loss50 condition in FCZ, CZ, CPZ electrodes. FCZ51, CZ51, CPZ51: win100 condition in FCZ, CZ, CPZ electrodes. ^a^ Bonferroni method: adjustments to post hoc multiple comparisons. * *p* < 0.05; ** *p* < 0.01. Abbreviations: ANOVA, repeated-measures analysis of variance; SD, standard deviation.

## Data Availability

The data that support the findings of this study are available from the corresponding author upon reasonable request.

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
