# Peer review of "High-Definition Transcranial Direct Current Stimulation Improves Decision-Making Ability: A Study Based on EEG"

_brainsci, 2023, doi:10.3390/brainsci13040640_

Round 1

Reviewer 1 Report

The study is an interesting evaluation of the effects of tDCS modulation on the left DLPFC as regards decision-making abilities. The study was performed on general population, with 2 types of stimulations: real and sham. I do not have specific concerns but there are some aspects that needed to be clarify. 

- the sample size is small. Have you evaluated the power analysis before the study? Otherwise, might you include a sensitivity analysis?

- please, include the approach applied for the randomization 

- How was performed the recruitment? Were participants naive about tDCS? Otherwise sham is detectable by users with previous exposure.

- You performed a lot of analysis on your data. Have you considered the possibility to adjust the p-values for multiple comparisons?

- It is not clear to me if your study is just a replication of previous studies or if there are some innovative aspects reported. Please, clarify this side, even because it might be a small replication.

Author Response

Point 1: the sample size is small. Have you evaluated the power analysis before the study? Otherwise, might you include a sensitivity analysis?

Response: In present study, the Bonferroni method was used to correct multiple comparisons in post hoc tests. In addition, partial eta squared(ηp2) values were obtained to examine the sizes of effects in the ANOVA models, where 0.05 indicated a small effect, 0.1 indicated a medium effect, and 0.2 indicated a large effect.

Point 2: Please, include the approach applied for the randomization. 

Response: In present study, we have described the approach applied for the randomization in line 113. Participants who meet inclusion criteria were randomly assigned in active group and sham group according to a computer-generated list.

Point 3: How was performed the recruitment? Were participants naive about tDCS? Otherwise sham is detectable by users with previous exposure.

Response: We are sorry for not describing the procedure of recruitmentment. All participants were recruited by advertisement in Anhui Medical University and colleges around it. Only the participants meet the criteria for enrollment, were included in the analysis.

All participants were naive of tDCS and other experiments. We have added “In addition, none of subjects participant in tES or TMS studies” in line 135. And the feasibility of the procedure were evaluated by asked the sensation in every HD-tDCS sessions. All participants were asked if they could guess what type of HD-tDCS they received after the study.The blinding efficacy of the procedure were We suggest that our blinding procedure was successful for participants were unable to distinguish between active and sham stimulation (active group = 56.25 %; sham group = 41.17 %; c² = 1.273, p = 0.529).

Point 4: You performed a lot of analysis on your data. Have you considered the possibility to adjust the p-values for multiple comparisons?

Response: The P values presented in this study were already corrected by Bonferroni method. We should apologize for not writting it in Statistical analysis section. 

Point 5: It is not clear to me if your study is just a replication of previous studies or if there are some innovative aspects reported. Please, clarify this side, even because it might be a small replication.

Response: Previous studies showed that left DLPFC participated in the decision making process. In present stusy, left DLPFC play a important role in decision making and anodal HD-tDCS is a promising technology in improving decision making. In addition, this study combined EEG and HD-tDCS, explained the mechanism of decision-making from an electrophysiological perspective in first time.

Reviewer 2 Report

The present research article by Zhou and colleagues, entitled ‘High-definition transcranial direct current stimulation improves decision making ability: A study based on EEG’ is a well-written and useful summary on the status of knowledge of the modulation effect of anodal HD-tDCS in decision making function. Authors reported that the improved decision making performance was associated with the increased power of theta oscillation. Furthermore, this study supports that theta oscillation may be involved in neuromodulation mechanism of decision making.

In general, I think the idea of this article is really interesting and the authors’ fascinating observations on this timely topic may be of interest to the readers of Brain Sciences. However, some comments, as well as some crucial evidence that should be included to support the author’s argumentation, needed to be addressed to improve the quality of the manuscript, its adequacy, and its readability prior to the publication in the present form, in particular reshaping parts of the Introduction and Methods sections by adding more evidence and theoretical constructs.

Please consider the following comments:

A graphical abstract that will visually summarize the main findings of the manuscript is highly recommended.

Abstract: According to the Journal’s guidelines, the abstract should be a total of about 200 words maximum. Please correct the actual one.

Please provide the full names before using abbreviations.

In general, I recommend authors to use more references to back their claims, especially in the Introduction of this article, which I believe is lacking. Thus, I recommend the authors to attempt to expand the topic of their article, as the bibliography is too concise. Nevertheless, I believe that less than 60/70 articles are too low for a research article. Therefore, I suggest the authors to focus their efforts on researching relevant literature: in my opinion, adding more citations will help to provide better and more accurate background to this study. 

Electrophysiological Recording: Please indicate how many component were removed using the Independent Component Analysis for artifact removal.

Introduction: The introduction would benefit from a reorganization of the sub-paragraphs. As it stands, there is confusion in terms of the flow of information. I suggest to begin with a theoretical explanation of techniques and protocols of neuromodulation techniques for those non expert readers (https://doi.org/10.1016/j.cub.2020.06.091; https://doi.org/10.1016/j.cortex.2021.01.004), with a special focus on how the most common methods of non-invasive brain stimulation techniques, which are proved to be efficient, effective, and well-tolerated for the study of specific cognitive functions. Moreover, I believe that it could be useful to focus on tDCS mechanisms of functions, the after-effects induced on neuroplasticity and current applications alongside (or in replacement of) drug treatments to speed recovery and improve motor and cognitive performance.

High-Definition Transcranial Direct Current Stimulation: This paragraph that describes stimulation parameters is the most important part of the study and should clearly describe all the experimental sessions in detail; therefore, this section might be improved including further explanations, allowing the effective communication of experimental procedures, like electrodes size and duration of stimulation. 

In my opinion, I think the ‘Conclusions’ paragraph would benefit from some thoughtful as well as in-depth considerations by the authors, because as it stands, it is very descriptive but not enough theoretical as a discussion should be. Authors should make an effort, trying to explain the theoretical implication as well as the translational application of their research.

Figures: I would suggest to modify all figures for clarity because, as it stands, the readers may have difficulty comprehending it and to change the scale of the vertical axis and use the same minimum/maximum scale value in all the graphs.

I hope that, after these careful revisions, this paper can meet the Journal’s high standards. 

I am available for a new round of revision of this paper. I declare no conflict of interest regarding this manuscript. 

Best regards,

Reviewer

Author Response

Point 1: A graphical abstract that will visually summarize the main findings of the manuscript is highly recommended.

Response: Thank you for your suggestion. We have uploaded the graphical abstract of the article.

Point 2: Abstract: According to the Journal’s guidelines, the abstract should be a total of about 200 words maximum. Please correct the actual one.

Response: We should apologize for ignoring the word count requirement for the abstract. We have correct the abstract in 187 words.

Point 3: Please provide the full names before using abbreviations.

Response: Thanks for your suggestion. We ensured that all the abbreviations used provided the full name.

Point 4: In general, I recommend authors to use more references to back their claims, especially in the Introduction of this article, which I believe is lacking. Thus, I recommend the authors to attempt to expand the topic of their article, as the bibliography is too concise. Nevertheless, I believe that less than 60/70 articles are too low for a research article. Therefore, I suggest the authors to focus their efforts on researching relevant literature: in my opinion, adding more citations will help to provide better and more accurate background to this study. 

Response: Thank you for your suggestion. We have revised the introduction section.

Ponit 5: Electrophysiological Recording: Please indicate how many component were removed using the Independent Component Analysis for artifact removal.

Response: Before HD-tDCS, the mean rejected epochs was 3.47 [95%, (2.68, 4.27)] in active group , 3.56 [95%, (2.57, 4.55)] in sham group. After HD-tDCS,the mean rejected epochs 3.76 [95%, (2.79, 4.74)] in active group and 3.50 [95%, (2.51, 4.49)] in sham group. Rejection epochs did not differ significantly among groups (F = 0.482, p = 0.493).

Ponit 6: Introduction: The introduction would benefit from a reorganization of the sub-paragraphs. As it stands, there is confusion in terms of the flow of information. I suggest to begin with a theoretical explanation of techniques and protocols of neuromodulation techniques for those non expert readers with a special focus on how the most common methods of non-invasive brain stimulation techniques, which are proved to be efficient, effective, and well-tolerated for the study of specific cognitive functions. Moreover, I believe that it could be useful to focus on tDCS mechanisms of functions, the after-effects induced on neuroplasticity and current applications alongside (or in replacement of) drug treatments to speed recovery and improve motor and cognitive performance.

Response: Thank you for your constructive suggestions. In this study, we are focus on the risk decision making under ambiguous condition. We have reorganization introduction part according to the Reviewer’s suggestion and highlighted it in yellow color.

Ponit 7: High-Definition Transcranial Direct Current Stimulation: This paragraph that describes stimulation parameters is the most important part of the study and should clearly describe all the experimental sessions in detail; therefore, this section might be improved including further explanations, allowing the effective communication of experimental procedures, like electrodes size and duration of stimulation. 

Response: We are sorry for not explain the stimulation parameters of HD-tDCS in detail. In present study, a battery-driven constant current stimulator (Neuroelectrics, Barcelona, Spain) was used to delivered current. Five Ag-AgCl sintered ring electrodes were held in plastic casings filled with conductive gel, embedded in an EEG cap, and attached to the adaptor device. Based on the 10-20 EEG system, the central electrode as anodal electrode was located in F3 (2 mA), and the remaining four return electrodes as cathodal electrode were located in F1/F5/FC3/AF3 (0.5 mA each). Sham HD-tDCS used the same electrode montage as the anodal stimulation. Anodal HD-tDCS was applied at 2 mA for 30 minutes, with 30 s ramped up and 30 s ramped down. Previous studies have shown that such intensity and total charge were safe for humans and sufficient to influence dlPFC activities [34-35]. The sham HD-tDCS stimulation duration was 60 s, with current fading in and out, which was similar to that of anodal HD-tDCS. The sham group subjects had the same HD-tDCS regimen and duration as those of the active group, while no current was given.

Ponit 8: In my opinion, I think the ‘Conclusions’ paragraph would benefit from some thoughtful as well as in-depth considerations by the authors, because as it stands, it is very descriptive but not enough theoretical as a discussion should be. Authors should make an effort, trying to explain the theoretical implication as well as the translational application of their research.

Response:Thang you for your suggestion. These finding supported that HD-tDCS is a appropriate technology protocol for facilitating the decision making process for those who are diagnosed with decision making dysfunction disorder. In addition, the approach of tDCS combined with EEG provides a direction for exploring the electrophysiological mechanisms in cognition. We have revised the conclusion section and highlighted in yellow color for it. 

Point 9: Figures: I would suggest to modify all figures for clarity because, as it stands, the readers may have difficulty comprehending it and to change the scale of the vertical axis and use the same minimum/maximum scale value in all the graphs.

 Response: Thank you for your suggestion. According to your suggestion, we have changed the scale of the vertical axis and use the same scale value in Figure 4 and Figure 5. The scale of the vertical axis of Figure 3B cannot use the same range for the different title unit.

Reviewer 3 Report

Authors proved HD-tDCS with healthy people. The analysis and experiments are in detail. The experiment setup with EEG looks good. There are no English grammar problems at all. Overall, the manuscript is well written so I recommend the manuscript could be minor if authors follow the suggstive guidelines as below.

1. Figure 1 and 4 quality need to be improved because the fonts are not clear to be seen.

2. Figures 3 and 4 fonts are too small to be seen.

3. No data availability section. 

4. In ref. please use abbreviated journal names.

5. Please check MDPI format for Author contribution section.

6. Please show the brief future work of the proposed research in Conclusion section.

7.  Please cite the sentence (During the recording, an electrode located at the forehead ~) with ref. (https://www.mdpi.com/1424-8220/22/16/6042).

8. How to obtain the resistance (All electrodes’ scalp resistance were kept below 5 kΩ.) ?

9. Please change Fig. 3B to Figure 3b. Please check others  according to MDPI format.

10. What is the unit for Factor time in Table 2 ? Is it min or sec ?

11. How to determine the score values in Figure 3.

12. Please use normal font for Figure label according to MDPI format.

13. What is the unit in x- and y-axes of Figure 6 ?

14. Please make a space between the last word and reference number. For example, connections[41] ->connections [41] in Line 386. 

15. Please do not use underbar and blue color in the Correspondence in Line 14.

Author Response

Point 1: Figure 1 and 4 quality need to be improved because the fonts are not clear to be seen.

Response: We are very sorry for this. We have improved the quality of Figure 1 and 4.

Point 2: Figures 3 and 4 fonts are too small to be seen.

Response: We are very sorry for using small font of Figures 3 and 4. We have used bigger font of Figures 3 and 4 for higher quality.

Point 3: No data availability section. 

Response: We are sorry for the ignorance. We have add data availability section in line 532.

Point 4: In ref. please use abbreviated journal names.

Response: Thank you for your suggestion. We have used correct format in ref.

Point 5: Please check MDPI format for Author contribution section.

Response: We are sorry for writing the error format for Author contribution section. We have modified the Author contribution section in the correct format.

Point 6: Please show the brief future work of the proposed research in Conclusion section.

Response: Thank you for your suggestion. In present study, anodal HD-tDCS over left DLPFC could improve decision making. So, it is reasonable to suppose that HD-tDCS is a appropriate protocol for facilitating the decision making process for those who are diagnosed with decision-deficit-related diseases. In addition, the approach of transcranial direct current stimulation combined with EEG provides a direction for exploring the electrophysiological mechanisms in cognition.

Point 7: Please cite the sentence (During the recording, an electrode located at the forehead ~) with ref. (https://www.mdpi.com/1424-8220/22/16/6042).

Response: Thank you for your suggestion. We have cited the article as the reference article.

Point 8: How to obtain the resistance (All electrodes’ scalp resistance were kept below 5 kΩ.)

Response: We are sorry for not describing the preparation process in details. Before electroencephalogram (EEG) recording, all participants were asked to wash their hair and keep their scalp dry. During the whole EEG recording period, the collection environment was kept quiet, the chamber with suitable temperature and light, and with a sound resistant. And before EEG recording, the computer showed the resistance of all electrode. Data acquisition began after all electrode’ resistances were below 5 kΩ.

Point 9: Please change Fig. 3B to Figure 3b. Please check others according to MDPI format.

Response: Thank you for your suggestion. We have examined the full text in detail and modified the language that does not meet the MDPI format.

Point 10: What is the unit for Factor time in Table 2 ? Is it min or sec ?

Response: We are sorry for the puzzled phrase. The Factor time which refers to the between-subject factor (pre-tDCS, post-tDCS). So, there is no unit of Factor time in Table 2 and Table 3.

Point 11: How to determine the score values in Figure 3.

Response: The score in Figure 3A is the netscores of each block which was defined as the number of low-risk choice minus the number of high-risk choice in each block. The score in Figure 3B is the total netscores which was defined as the number of low-risk choice minus the number of high-risk choice. To avoid confusion, we replaced the scores and gave a detailed description of the definition of the scores.

Point 12: Please use normal font for Figure label according to MDPI format.

Response: We have used the normal font Figure label according to MDPI format. The revised section were highlighted in yellow color.

Point 13: What is the unit in x- and y-axes of Figure 6 ?

Response: The x-axis represents the changes in the ratio of the negative feedback. So, there is no unit in x-axes of Figure 6. The y-axes represents the changes in theta band power. So, the unit in y-axes of Figure 6 is dB. We have add the unit in x-axes of Figure 6.  

Point 14: Please make a space between the last word and reference number. For example, connections[41] ->connections [41] in Line 386. 

Response: We are sorry to place the reference serial number in the wrong position. We have placed the cited references in the correct position.

Point 15: Please do not use underbar and blue color in the Correspondence in Line 14.

Response:We are very sorry to underbar and blue color in the Correspondence. We have deleted the underbar in the Correspondence and wrote the Corrspondence in black color.

Reviewer 4 Report

Zhou et al, conducted a basic study that included exploring the effect of multiple stimulation sessions on decision making as measured by the IGT.   The study measured both behavioral and EEG measures to evaluate the stimulation effects.
  The amount of stimulation was massive (10 30-minutes sessions of 2mA) and teh montage included only the left hemisphere, proper reports of side effects including pain, burning and itching should be added and compared between the two conditions, as this amount is far larger than what is usually conducted with healthy young subjects. Electrode sizes are not mentioned, but if it is small electrodes, current density is well above safety levels, see Bikson, M., Grossman, P., Thomas, C., Zannou, A. L., Jiang, J., Adnan, T., ... & Woods, A. J. (2016). Safety of transcranial direct current stimulation: evidence based update 2016. Brain stimulation9(5), 641-661.   The activation of teh left hemipshere suggests that responses should be analyzed according to visual field of stimuli presentation (the choice between 50 and 100 coins). ‏   Moca is a weird choice for a young and healthy sample, they should all be above 29 points, and not show changes in two weeks.  There was nearly a 2 years difference between the groups (with z of 1.4 (why not proper t?)) with older subjects in teh active stimulation group, so how comes their education level was equal ?    Many decision making research think evaluation of choices takes place in VMPFC networks (see for example Baumgartner et al 2011), so why stimulating the DLPFC? requires further justification. if stimulation conditions include only one anodal and one sham condition, there is no way to conclude that this is anything to do with the DLPFC specifically, could be just general stimulation effect (the subjects in teh active condition were angry at the end of teh 10 painfull sessions, for example) Baumgartner, T., Knoch, D., Hotz, P., Eisenegger, C., & Fehr, E. (2011). Dorsolateral and ventromedial prefrontal cortex orchestrate normative choice. Nature neuroscience14(11), 1468-1474.   I see Tables 2 & 3 and their related analyses as descriptives only and not well justified, as the analyses did not address the heavy corelations and dependencies between the various repetaed measures.   There is a philosophical issue here - whether increased risk taking is indeed a better deciiosn making? perhaps it is due to the lower activation in other left hemipshere hubs rather than the anodal activation of teh DLPF? pity more DM tasks were not used, considering teh vast amount of stimulation.

Author Response

Point 1: The amount of stimulation was massive (10 30-minutes sessions of 2mA) and teh montage included only the left hemisphere, proper reports of side effects including pain, burning and itching should be added and compared between the two conditions, as this amount is far larger than what is usually conducted with healthy young subjects. Electrode sizes are not mentioned, but if it is small electrodes, current density is well above safety levels, see Bikson, M., Grossman, P., Thomas, C., Zannou, A. L., Jiang, J., Adnan, T., ... & Woods, A. J. (2016). Safety of transcranial direct current stimulation: evidence based update 2016. Brain stimulation, 9(5), 641-661.   

Response: Thank you for your suggestion. We have described the side effect in Result section in line 451. Five participants reported tingling and burning sensation, three participants reported itching sensation under the electrodes during the first 30 s of stimulation(c2 =0.339, p=0.560). And we have add the size of electrode according to reviewer 2. The five Ag-AgCl with a diameter of 12 mm sintered ring electrodes were held in plastic casings filled with conductive gel, embedded in an EEG cap, and attached to the adaptor device. Hence, the HD-tDCS with 2 mA, 30 minutes is safe for healthy participants.

Point 2: The activation of teh left hemisphere suggests that responses should be analyzed according to visual field of stimuli presentation (the choice between 50 and 100 coins). ‏  

Response: Thank you for your suggestion. Do you mean that the location of the “50” and “100” would affect the result? According to this, there are some reasons to eliminate your confusion. Firstly, the classical IGT was the same location of advantage and disadvantage choice (advantage deck presented in left of computer’ screen and disadvantage deck presented in right of computer’ screen). Secondly, there are several studies have proved that the revised IGT could detect the decision making dysfunction disorders. Thirdly, event-related functional magnetic resonance imaging studies showed that a neural circuitry involving the left DLPFC was activated during the IGT and that the changed activity of left DLPFC was associated with the IGT netscores [1]. Patients with DLPFC impairment had poor decision making performance [2], especially when the damage in the left side [3]. Therefore, IGT is a promising neuropsychological measurement paradigm for measuring decision making function.  

Point 3: Moca is a weird choice for a young and healthy sample, they should all be above 29 points, and not show changes in two weeks. 

Response: Thank you for your suggestion. We re-checked the raw data of MoCA, and the MoCA data were indeed correct. The changed MoCA score after stimulation may be the learning effect of MoCA.

Point 4: There was nearly a 2 years difference between the groups (with z of 1.4 (why not proper t?)) with older subjects in teh active stimulation group, so how comes their education level was equal ? 

Response: Thank you for your suggestion. We should apologize for the error method in year, education. We have used Mann-Whitney U test to compare the difference between two groups. We have revised the data of Table1 and add the relevant content in line 259.

Point 5: Many decision making research think evaluation of choices takes place in VMPFC networks (see for example Baumgartner et al 2011), so why stimulating the DLPFC? requires further justification.

Response: Neural circuits underlying decision making include cortical, subcortical and cerebellar nodes, with the prefrontal cortex (PFC) and its different functional subdivisions as a central processing hub [4]. In present study, we focused on left DLPFC for it established role in mediating risk decision making under ambiguous condition. Specifically, the functions of the left DLPFC are closely related to risk taking, ambiguous decisions, and intertemporal decisions [5]. The left DLPFC, the main region for cognitive control, coordinates the reward processing system and the risk processing system . Indeed, functional MRI [6], stimulation and lesion studies have associated the left DLPFC with risky decisions. Since an increase in avoid make risk decision behavior was observed after up-regulating the left DLPFC via anodal stimulation [7], suppressing activity in the left DLPFC (via cathodal stimulation) is expected to reduce risk disposition [8].

Point 6: if stimulation conditions include only one anodal and one sham condition, there is no way to conclude that this is anything to do with the DLPFC specifically, could be just general stimulation effect (the subjects in teh active condition were angry at the end of teh 10 painfull sessions, for example) Baumgartner, T., Knoch, D., Hotz, P., Eisenegger, C., & Fehr, E. (2011). Dorsolateral and ventromedial prefrontal cortex orchestrate normative choice. Nature neuroscience, 14(11), 1468-1474.   I see Tables 2 & 3 and their related analyses as descriptives only and not well justified, as the analyses did not address the heavy correlations and dependencies between the various repeated measures.   There is a philosophical issue here - whether increased risk taking is indeed a better decision making? perhaps it is due to the lower activation in other left hemisphere hubs rather than the anodal activation of teh DLPF? pity more DM tasks were not used, considering teh vast amount of stimulation.

Response:Thank you for your suggestion. Firstly, none of participants reported serious side effect, like headache or emotional change. Secondly, we should apologize for not writing the statistics section in details. In present study, the Bonferroni method was used to correct multiple comparisons in post hoc tests. So, it is reasonable to speculate that anodal HD-tDCS applied left DLPFC can improve decision making effectively. In addition, a better decision making performance may be associate with the loss aversion in this study. This study duplicated previous studies that anodal tDCS is a promising technology in improving decision making by targeting the left DLPFC. The innovation of this study is in exploring the electrophysiological mechanisms of modulate effect by HD-tDCS. 

Round 2

Reviewer 1 Report

I think the authors have addressed my concerns and the manuscript might be accepted

Reviewer 2 Report

The authors did an excellent job clarifying all the questions I have raised in my previous round of review. Currently, this paper is a well-written, timely piece of research that described the modulation effect of anodal HD-tDCS in decision making function-

Overall, this is a timely and needed work. It is well researched and nicely written, therefore I believe that this paper does not need a further revision, therefore the manuscript meets the Journal’s high standards for publication.

I am always available for other reviews of such interesting and important articles.

Thank You for your work, Reviewer